# Diet Composition Affects Liver and Mammary Tissue Transcriptome in Primiparous Holstein Dairy Cows

**DOI:** 10.3390/ani10071191

**Published:** 2020-07-14

**Authors:** Shengtao Gao, Zheng Zhou, Jiaqi Wang, Juan Loor, Massimo Bionaz, Lu Ma, Dengpan Bu

**Affiliations:** 1State Key Laboratory of Animal Nutrition, Institute of Animal Science, Chinese Academy of Agricultural Sciences, Beijing 100193, China; gaoshengtao1990@163.com (S.G.); wangjiaqi@caas.cn (J.W.); 2Department of Animal Science, Michigan State University, East Lansing, MI 48824, USA; zhouzz@msu.edu; 3Department of Animal Sciences and Division of Nutritional Sciences, University of Illinois, Urbana, IL 17019, USA; jloor@illinois.edu; 4Animal and Rangeland Sciences, Oregon State University, Corvallis, OR 97331, USA; massimo.bionaz@oregonstate.edu; 5Joint Laboratory on Integrated Crop-Tree-Livestock Systems of the Chinese Academy of Agricultural Sciences (CAAS), Ethiopian Institute of Agricultural Research (EIAR) and World Agroforestry Center (ICRAF), Beijing 100193, China

**Keywords:** forage-to-concentrate ratio, liver and mammary gland, transcriptome, corn stover

## Abstract

**Simple Summary:**

Corn stover (CS) diets are still adopted by small dairy farms in China. Compared with mixed forage (MF) diets, feeding CS diet affects milk composition, digestibility, feed intake, ruminal fermentation, and lactation performance. Mammary gland and liver are two of the most important organs for lactation in cows. However, research related to the effect of CS diet on the metabolism of liver and mammary tissues of dairy cows is limited. Overall evaluation of the biological response of liver and mammary tissues of dairy cows to changes in CS diet compared with MF diet is essential. Thus, the objective of the present study was to evaluate the overall adaptation of liver and mammary tissue to a CS diet in mid-lactation primiparous dairy cows. Modest effect on the transcriptome of the liver and mammary tissue by the CS diet was observed. The analysis of the genes affected by CS indicated mammary gland responding to lower level of linoleate from the diet (lower in CS vs. MF) by activating the associated biosynthesis metabolism while the liver adaptively activated potassium transport to compensate for a lower K ingestion. The results of this study may facilitate the development of better feeding and management strategies and the increase of profitability of dairy farms.

**Abstract:**

The objective of the present study was to evaluate the overall adaptations of liver and mammary tissue to a corn stover (CS) compared to a mixed forage (MF) diet in mid-lactation primiparous dairy cows. Twenty-four primiparous lactating Holstein cows were randomly allocated to 2 groups receiving either an alfalfa forage diet (MF, F:C = 60:40) with Chinese wildrye, alfalfa hay and corn silage as forage source or a corn stover forage diet (CS, F:C = 40:60). A subgroup of cows (n = 5/diet) was used for analysis of liver and mammary transcriptome using a 4 × 44K Bovine Agilent microarray chip. The results of functional annotation analysis showed that in liver CS vs. MF inhibited pathways related to lipid metabolism while induced the activity of the potassium channel. In mammary tissue, fatty acid metabolism was activated in CS vs. MF. In conclusion, the analysis of genes affected by CS vs. MF indicated mammary gland responding to lower level of linoleate from the diet (lower in CS vs. MF) by activating the associated biosynthesis metabolic pathway while the liver adaptively activated potassium transport to compensate for a lower K ingestion.

## 1. Introduction

Mammary gland and liver are two of the most important organs for lactation in cows. The mammary gland has an incredible level of organization and a remarkable ability to convert circulating nutrients (amino acids, glucose, and fatty acids) into milk components [1]. The productivity of this biological factory is extensive, and mammary epithelial cells would rank second only to the photosynthetic cells as a factor in sustaining mammalian life [1]. The high anabolic status of the mammary gland during lactation drives almost 5-fold increase in energy and protein requirements from late gestation to lactation in dairy cows [2]; thus, mammary gland is highly dependent from the availability of nutrients. Liver plays a central role in supporting the anabolic status of the mammary gland. In ruminants, >90% of the glucose available for all functions is produced de novo. Liver contributes to >80% of the glucose produced via gluconeogenesis, as demonstrated in sheep [3,4]. Therefore, the large amount of glucose used for synthesis of milk lactose is coming largely from the hepatic gluconeogenesis. In addition, the liver plays a central role in lipid metabolism, amino acid metabolism, detoxification, and immune defense [5,6]. 

It is estimated that more than 10 million tons of corn stover are generated annually in China [7]. Taking full advantage of crop residues such as corn stover (CS) to meet the demand of forage and reduce dependence on imported alfalfa hay has been a strategic policy for the Chinese dairy industry [8]. Corn stover diets are still adopted by small dairy farms in China [9]. Comparison between corn stover diet and alfalfa diet has been conducted in previous studies [8,10,11,12]. Several studies demonstrated that milk performance was decreased for dairy cows under a CS diet [8,13,14]. In addition, prior studies revelated alteration of microbial protein synthesis, N utilization efficiency [8], and rumen microbiota of dairy cows [11] induced by CS diet compared with alfalfa diet in dairy cows. Thus, it is expected that the two main organs involved in milk synthesis in dairy cows, i.e., liver and mammary, would be affected by feeding CS instead of alfalfa diets to dairy cows. Thus, the objective of the present study was to evaluate the overall adaptation of liver and mammary tissue to a CS diet compared with alfalfa diet via transcriptome analysis in mid-lactation primiparous dairy cows. 

## 2. Materials and Methods 

### 2.1. Ethics Statement

All animal care and procedures were approved by the Animal Welfare and Ethical Committee of Institute of Animal Science, Chinese Academy of Agricultural Sciences (No. IAS20180115). The whole experiment was conducted in strict accordance with the Directions for Caring of Experimental Animals from the Institute of Animal Science, Chinese Academy of Agricultural Sciences.

### 2.2. Experimental Design

The present study was a sub study of a larger experiment, and the design has been described previously by Han et al. (2014) [14]. Briefly, 24 primiparous lactating Holstein cows (body weight, 558 ± 10 kg; days in milk, 136 ± 37; daily milk yield, 21.12 ± 2.30 kg) were assigned to two groups. The cows in group 1 were fed with an alfalfa diet containing Chinese wild rye, alfalfa hay, and corn silage as forage sources (Mixed Forage (MF); F:C = 60:40, *n* = 12), and cows in group 2 were fed with a corn stover diet with corn stover as forage source (CS, F:C = 40:60, *n* = 12). The DM (55.13 g/100 g of DM]), CP (16.80 g/100 g of DM), NDF (42.60 g/100 g of DM), and NEL content (6.27 MJ/kg of DM) of the two diets were similar (Appendix A). The diets were mixed and supplied to the cows twice per day at 07:00 and 19:00 h and allowed for 5–10% orts. The cows were milked twice daily at 07:00 and 19:00 h. Cows received ad libitum feed and water. The ingredients and the chemical composition of the two experimental diets are reported in Han et al. (2014) [14]. The total duration of the experiment was a single period of 11 weeks, including an adaptation period of 2 weeks before the experimental period of 9 weeks. 

### 2.3. Tissue Collection

Five cows from each treatment group, without mastitis, pregnancy, and in their first lactation, were used for liver and mammary gland biopsies at the end of 9-week feeding trial. The liver and mammary tissue biopsies were performed simultaneously (i.e., within 40 min) at approximately 0700 h after the AM milking. Biopsy procedure was conducted as previously described [15]. Briefly, the mammary biopsy was performed on the midsection of left rear quarter, and the liver biopsy was done between the 11th and 12th rib on the right side of cows. Before the biopsies, a small dose of xylazine (0.05 mg/kg BW) was injected intramuscularly before intramuscular injection of 3 to 4 mL of lidocaine-hydrochloride (2% solution) as local anesthetic. The biopsy of mammary gland was performed using a cordless drill equipped with a bioptic probe (AgResearch Ruakura, Ruakura Agricultural Center, Hamilton, New Zealand, 85 mm in length by 4.5 mm in diameter), and the biopsy of liver was performed using a Tru-Cut biopsy needle (Tru-Cut Biopsy Needle, Baxter Healthcare Corp., Valencia, CA, USA, diameter 4 mm). Approx. 400 mg of mammary tissue and 300 mg of liver tissue were obtained from the biopsy. For both biopsies, 4 or 5 Michel clips (11 mm; Henry Schein, Melville, NY, USA) were used to close the skin incision. Tissue samples were stored in liquid nitrogen immediately after rinsing with PBS buffer prepared with RNase, DNase-free water. 

### 2.4. RNA Extraction and Microarray Analysis

Total RNA of each sample was isolated using TRIzol reagent (Life technologies, US, Cat#74106) according to the manufacturer’s protocol. Rneasy mini kit (QIAGEN, Germany, Cat#74106) and RNase-Free DNase Set (QIAGEN, Germany, Cat#79254) were used to purify the total RNA. NanoDrop1000 was used to measure the concentration of the total RNA. 2100 Bioanalyzer (Agilent Technologies, US) and the RNA 6000 Nano Kit (Agilent Technologies, US) were used to assess the integrity of the purified total RNA. The OD260/OD280 values were ≥1.9 and the RIN (RNA Integrity Number) values were ≥ 8.0. 

The transcriptomic analysis processing has been described in Bu et al. (2017) [15]. Briefly, a 4 × 44 K Bovine microarray chip (Agilent Technologies, US, design ID: 023647) with the capacity to measure 17,252 unique annotated genes was used for the transcriptomic analysis. Firstly, we used Low Input Quick Amp Labeling Kit, One-Color (Agilent technologies, US Cat#5190 ± 2305) according to the manufacturer’s instructions to amplify and label the RNA. Then we hybridized the slides with 1.65 μg Cy3-labeled cRNA using Gene Expression Hybridization Kit (Agilent technologies, US, Cat#5188 ± 5242,) in a hybridization oven (Agilent technologies, US). Finally, we washed the slides in staining dishes (Thermo Shandon, US) with Gene Expression Wash Buffer Kit (Agilent technologies, US, Cat#5188 ± 5327) after 17 h hybridization. 

Agilent Microarray Scanner (Agilent technologies, US) was used to scan the slides with default settings (i.e., dye channel: Green, Scan resolution = 5 μm, PMT, 100%, 10%, 16bit), and with Feature Extraction software 10.7 (Agilent technologies, Santa Clara, CA, US). The microarray dataset presented in this manuscript was deposited at NCBI’s Gene Expression Omnibus and is accessible through GEO Series accession number GSE73980.

### 2.5. Statistical Analysis

#### 2.5.1. Differentially Expressed Genes (DEG) Analysis

Raw data were normalized by quantile method with Gene Spring Software 11.0 (Agilent technologies, US) and uploaded into JMP Genomics (SAS INSTITUTE, Cary, NC, USA) for statistical analysis using ANOVA with diet, tissue, and their interaction as main effect and cows as random effect. Data were log_2_ transformed and values of the annotated genes with multiple oligos were averaged before statistical analysis [15]. A false discovery rate (FDR) correction was applied and a FDR < 0.2 was used as cut off for identification of DEG. The complete datasets with DEG is available in Appendix A.

#### 2.5.2. Functional Annotation Analysis of Differently Expressed Genes

Kyoto Encyclopedia of Genes and Genomes (KEGG) pathways was mainly analyzed with the Dynamic Impact Approach (DIA) [16] as previously described [15,17]. We uploaded the dataset with Entrez Gene ID, FDR-adjusted p-value, expression ratio, and P-value to DIA and Entrez Gene ID of the microarray were used as background. FDR ≤ 0.2 and *p*-value ≤ 0.05 between CS and MF were used as cut-off. The enrichment analysis for the DEG was also conducted and visualized using ClueGO [18] and Database for Annotation, Visualization, and Integrated Discovery (DAVID) [19]. For these analyses, the up-regulated DEG and down-regulated DEGs were uploaded separately to DAVID and ClueGO. The results of DAVID analysis were downloaded using the Functional Annotation Chart. For ClueGO analysis, we selected “GO Biological Process” with evidence code “All”, and selected the option “Show only Pathways with p…” and keep default value of 0.05. The visualized results were exported as PDF format and transformed to pictures using Adobe Illustrator CC 2019. 

## 3. Results and Discussion

The cows used in the present study were a subset of a larger study [14]. In that study, the different diet (CS vs. MF) did not affect milk yield and milk fat content, whereas dry matter intake (*p <* 0.01) and milk fat yield (*p <* 0.05) were higher in MF compared with CS [14]. To evaluate the influence of CS diet on the biology of liver and mammary gland in dairy cows, we collected mammary and liver tissues, extracted the total RNA, and measured the transcriptome using microarray. The results uncovered 139 DEG affected in liver, 120 DEG affected in mammary gland, and 292 DEG overall affected in liver and mammary gland by CS vs. MF (*p <* 0.05, FDR < 0.2) (Figure 1). The small number of DEG indicate a relatively modest effect of CS diet on the transcriptome of liver and mammary tissue. 

### 3.1. Signaling Pathways in Liver and Mammary Tissues are Affected by CS vs. MF

The bioinformatic analysis of DEGs in liver and mammary tissues affected by CS vs. MF diet performed using DIA revealed an overall low impact on pathways with the largest impact detected in metabolic-related pathways (Figure 2; Appendix A). In particular, pathways associated with carbohydrate, lipid, and glycan biosynthesis metabolism were among the most impacted. 

When considering the overall effect of CS vs. MF, the “Glycosphingolipid biosynthesis—ganglio series” was the most affected pathway and was inhibited with CS vs. MF (Figure 3, CS/MF). This effect was mostly driven by the large effect of CS vs. MF in mammary tissue (Figure 3, CS/MF Mammary). Several signaling associated pathways (“Adipocytokine signaling pathway” and “NOD-like receptor signaling”) were inhibited in cows fed CS vs. cows fed MF (Figure 3, CS/MF). Adipocytokines, also called adipokines, are a series of signaling molecules including leptin, adiponectin, IL-1β, IL-6, and serum amyloid A produced and released by the adipose tissue and are involved in the inflammation, insulin resistance, and acute-phase responses [20]. One of the main outcomes of the activation of “NOD-like receptor signaling” is the activation of NF-κB [21]. NF-κB is found in almost all animal cell types and plays a key role in regulating the immune response to infection [22]. In this study, the inhibition of “Adipocytokine signaling pathway” and “NOD-like receptor signaling” indicated an overall downregulation of inflammation in liver and mammary tissues in cows fed CS diet compared to the cows fed MF diet. 

The results of DAVID analysis of DEG affected by CS vs. MF when considering both liver and mammary tissue are shown in Figure 4, CS/MF (complete DAVID results are available in Appendix A). The up-regulated DEG were enriched by KEGG pathways “cAMP signaling pathway” and “Neuroactive ligand-receptor interaction” and by the GO biological processes “meiotic DNA repair synthesis” and “proteolysis involved in cellular protein catabolic process”. The GO biological processes “Cellular response to UV” was the only enriched term among downregulated DEG. Cyclic adenosine 3′,5′-monophosphate (cAMP) is one of the most common and universal second messengers in cells. The cAMP pathway is induced by ligands including hormones, neurotransmitters, and other signaling molecules [23]. Neuroactive ligand-receptor interaction pathway is a collection of receptors and ligands associated with intracellular and extracellular signaling pathways on the plasma membrane [24]. According to KEGG, the cAMP pathway is integral to the “Neuroactive ligand–receptor interaction” pathway (Appendix A). cAMP plays a pivotal role in neuronal survival [25]. Thus, in this study, the concomitant induction of “cAMP signaling pathway” and the “Neuroactive ligand–receptor interaction” could indicate an increase in sensitivity of innervations in liver and mammary tissues in CS vs. MF pathway. Detail visualization of the “Neuroactive ligand–receptor interaction” pathway (Appendix A) revealed that the enrichment of this pathway was due to increase in expression in CS vs. MF of genes involved in the epinephrine, neuropeptides, glutamate ionotropic, and GABA receptors. Interestingly, epinephrine are known to inhibit oxytocin induction of milk let down, partly due also to an effect on the myoepithelial cells [26]. However, the same pathway was not enriched in up-regulated DEG in CS vs. MF in mammary tissue; thus, it is unclear the significance of the enrichment of the above pathways. 

Taken together, the bioinformatic analyses of the DEG affected by CS vs. MF in both liver and mammary tissue reveal an effect on few signaling pathways related to adipokines and neurons. Despite an indication of a higher sensitivity to various neuronal activators but a lower sensitivity to adipokines in CS vs. MF cows in liver and mammary tissue, the significance of the observed effects remains unclear. 

### 3.2. Liver: CS Diet Inhibits Metabolism of Lipid

Among lipid-related pathways in the results of DIA, the inhibition of steroid hormone biosynthesis and PPAR signaling and activation of bile acid biosynthesis were among the top impacted pathways (Figure 3, CS/MF Liver). Although we detected an inconsistent direction of the effects of the above three pathways associated with lipid metabolism, the whole lipid metabolism in liver can be considered inhibited by CS vs. MF as shown in Figure 2 (CS/MF Liver). The PPAR isotypes, including PPARα and PPARδ, are nuclear proteins that mediate the effects of NEFA on peroxisomal and mitochondrial fatty acid oxidation and upregulate genes associated with ketogenesis and ureagenesis [27]. The PPAR isotypes are considered important in the whole economy of lipid metabolism in liver of mammals [28]. Recent data indicated PPARδ being the major PPAR isotypes responding to NEFA in bovine liver [29]. Shahzad et al. (2014) demonstrated that “PPAR signaling” pathway plays a role in regulating the transcriptomic adaptation of the liver to different levels of dietary energy in prepartum cows [30]. Thus, in the present study, the modest inhibition of PPAR signaling pathway by CS compared to MF can be a response to different dietary energy levels between the two diets (lower in CS vs. MF) but also to the different composition of fatty acids in the diet, with CS having less 16:0 in the diet and less 18:0 intake in CS vs. MF, both with evidence of PPAR activation in dairy cows [29,31]. 

Steroid hormones including mineralocorticoids, glucocorticoids, androgens, estrogens, and progestogens [32,33] are involved in regulation of various biological pathways associated with the reproductive system and maintenance of the entire-body metabolic homeostasis [34]. The liver plays a crucial role in steroid hormones and general cholesterol metabolism, including biosynthesis of cholesterol [34,35]. The study of Shi et al. (2018) indicated that cholesterol synthesis in liver was quadratically increased with increasing dietary forage levels from 20% to 80%, with a peak detected with 60% forage-to-concentrate ratio [36]. Consistent with that study, in the present study, the pathway of “Steroid hormone biosynthesis” was inhibited by CS diet containing a low forage to concentrate ratio (40:60) compared to MF with a high forage to concentrate ratio (60:40) (Figure 3, CS/MF Liver). It is known that the synthesis of cholesterol is a highly energy-consuming process with 36 mol of ATP for 1 mol of cholesterol produced [37]. Cholesterol metabolism has a close relationship with whole-body energy partitioning [38,39]. Previous studies reported that during feed deprivation energy and glucose are reassigned to the lactating mammary gland by inhibiting the biosynthesis of cholesterol in the liver [40,41]. Both the energy content in CS diet and DMI of CS cows were all lower than MF diet (6.19 vs. 6.35 MJ/kg of DM; 18.8 vs. 20.3 kg/d, *p* < 0.01) [14]. Thus, the inhibition of steroid hormone biosynthesis in liver of CS vs. MF in this study appears to be a potential adaptive mechanism to divert energy and glucose to the mammary gland. 

### 3.3. Liver: CS Diet Activated Folate Biosynthesis and Homologous Recombination

Folate has antioxidant functions and lipid-lowering effects [42]. Liver has an important role in maintaining whole body folate homeostasis and is a major organ for folate storage and metabolism [43]. In addition, folate homeostasis is affected by reabsorption of bile folates via the enterohepatic circulation [44]. Interestingly, in this study, both the pathways of “Folate biosynthesis” and “Primary bile acid biosynthesis” were activated by CS vs. MF (Figure 3, CS/MF Liver), indicating an increased folate biosynthesis by liver of cows receiving CS compared with MF.

Data revealed that the “Homologous recombination” pathway was activated in CS vs. MF. Indicating an increase in replication and repair of the genome. Together with the above activation of folate biosynthesis, increased replication might imply an increased requirement of folic acid to synthesize nucleic acid [45]. Thus, the results of this study seemingly indicate a higher folate requirement in cows fed with CS diet compared to MF diet, while the activated folate biosynthesis in liver appears to aim to meet the requirement. 

### 3.4. Liver: CS Diet Induces Potassium Transport in Liver

The functional annotation results of DAVID and ClueGO suggested an activation of potassium channel activity in CS compared with MF (Figure 4 and Figure 5). The activated potassium channel activity implied an activated potassium transport ability of liver by CS compared with MF. It is known that corn stover is much lower in K than alfalfa hay, corn silage, and Chinese wild rye (0.9 vs. 2.37, 1.20, and 3.34 g/100 g of DM respectively) [46]. Furthermore, the DMI of CS was significantly lower than MF (18.8 kg vs. 20.3 kg, *p <* 0.01) [14]. Consequently, the K ingested by cows feed with CS diet were 61.6% lower than the cows feed with MF diet (190.7 g/d vs. 308.2 g/d, calculated using data from NRC (2001) [46]. Therefore, in the present study, the activation of potassium channel activity in liver of CS cows compared with MF cows may be induced by lower K ingestion, to furthest increase the absorption ability of K in liver. 

### 3.5. CS vs. MF Induces Lipid Metabolism in Mammary Tissue

In the mammary tissue, CS vs. MF had the largest impact on metabolic-associated pathways (Figure 2, CS/MF Mammary). All top impacted pathways by CS vs. MF are associated with “Metabolism” and “Organismal Systems” with the most impacted pathway being “Glycosphingolipid biosynthesis—ganglio series” which was inhibited (Figure 3, CS/MF Mammary). This pathway is among the most activated during lactation in dairy cows, as revealed by a large transcriptomic study [47], with still an unknown clear association with milk synthesis. In this study, the milk production of CS cows was only numerically lower than MF cows. 

The data of this study indicated that the fatty acid metabolism (especially the biosynthesis of alpha-linolenic acid and linolenic acid) in mammary gland was activated in cows receiving CS diet compared to MF diet (Figure 2 and Figure 3, CS/MF Mammary). Among the top 10 impacted pathways in DIA, both the pathways of “alpha-Linolenic acid metabolism” and “Linolenic acid metabolism” were activated (Figure 3, CS/MF Mammary). The importance of induced lipid synthesis-related pathways was also supported by results from DAVID and ClueGO (Figure 4 and Figure 5), where the pathways of “Arachidonic acid metabolism” and “Linolenic acid metabolism” were enriched by the upregulated DEG in CS vs. MF. The content of C18:3 was lower in milk of CS vs. MF (0.27 vs. 0.39 g/100g total fatty acids, *p* < 0.01; 0.83 vs. 0.94 kg/d, *p* < 0.05) [14]. The ingestion of C18:3 by CS cows was > 2.5-fold lower in CS vs. MF cows (12.60 vs. 32.43 g/d), while secretion of C18:3 fatty acid by CS cows was only 1.64-fold lower than MF cows (2.24 vs. 3.66 g/d). Furthermore, at around 120 day relative to parturition, almost 60% of fatty acids in milk are synthesized de novo in mammary gland [48]. The induction of pathways related to biosynthesis of fatty acids in CS vs. MF has likely compensated the lower availability of preformed 18:3. 

## 4. Conclusions

The CS diet compared with MF diet had very modest effect on the transcriptome of the liver and mammary tissue of primiparous cows. The bioinformatics analysis of the DEG affected by CS vs. MF on both liver and mammary tissues revealed an effect on several signaling pathways while the significance remains unclear. When the bioinformatics analysis was performed on DEG affected only in liver, results indicated a modest inhibition of PPAR signaling pathway and steroid hormone biosynthesis but induced folate biosynthesis, repair/duplication of the DNA, and transport of K in CS compared to MF. The DEG affected by CS vs. MF in mammary gland revealed an activation of linoleate metabolism. Taken together, results of this study implied that mammary gland can make up the deficiency of linoleate from diet by activating fatty acid biosynthesis and liver can adaptively activate the potassium transport under lower K ingestion.

## Figures and Tables

**Figure 1 animals-10-01191-f001:**
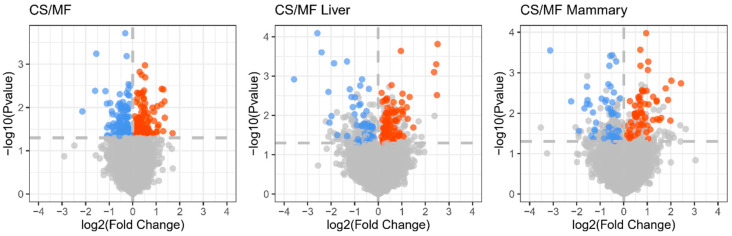
Volcano plot for the genes statistically (A false discovery rate (FDR) < 0.2) affected by corn stover (CS) vs. mixed forage (MF) diet in both liver and mammary tissue (CS/MF; 140 up- and 152 down-regulated), liver (CS/MF Liver; 56 down- and 83 up-regulated), and mammary tissue (CS/MF Mammary; 63 up-D and 57 down-regulated). Red and blue dots represent upregulated and downregulated DEG, respectively.

**Figure 2 animals-10-01191-f002:**
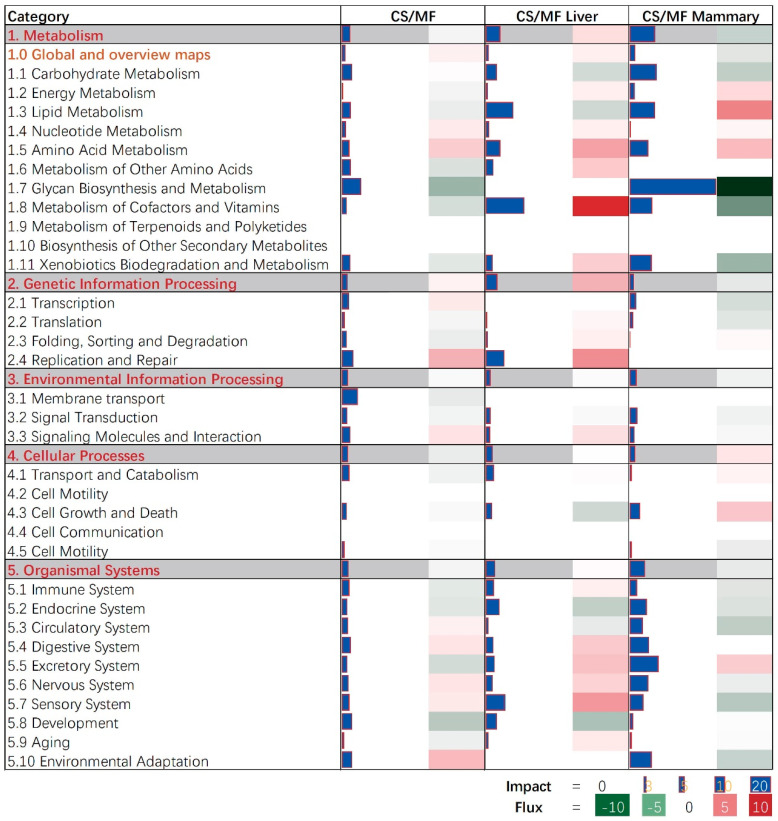
Summary of the main categories and sub-categories of Kyoto Encyclopedia of Genes and Genomes (KEGG) pathways as results of the transcriptomic effect on mammary (M) and liver (L) tissue of corn stover (CS) compared to mixed forage (MF) ration in dairy cows as analyzed by the Dynamic Impact Approach. On the right are the bar denoting the overall impact (in blue) and the shade denoting the effect on the pathway (from green—inhibited—to red—activated). Darker the color larger the activation (if red) or inhibition (if green) of the pathway.

**Figure 3 animals-10-01191-f003:**
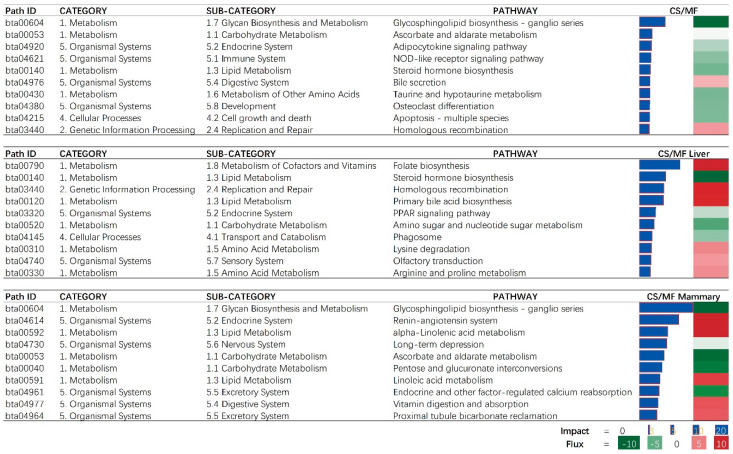
The 10 most impacted pathways in liver (CS/MF Liver), mammary (CS/MF Mammary), and overall of liver and mammary (CS/MF) of corn stover (CS) compared to mixed forage (MF) ration in dairy cows uncovered by the Dynamic Impact Approach. On the right are the bar denoting the overall impact (in blue) and the shade denoting the effect on the pathway (from green—inhibited—to red—activated). Darker the color larger the activation (if red) or inhibition (if green) of the pathway.

**Figure 4 animals-10-01191-f004:**
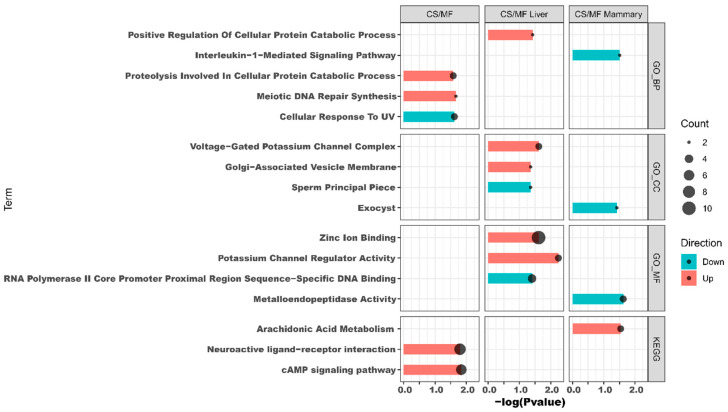
Functional analysis results revealed by Database for Annotation, Visualization, and Integrated Discovery (DAVID) analysis of the transcripts up- (in red shade in the figure) or down- (in blue shade in the figure) regulated by corn stover (CS) compared to mixed forage (MF) in both liver and mammary (CS/MF), in liver tissue alone (CS/MF Liver), or mammary tissue alone (CS/MF Mammary). GO_MF: Gene Ontology Molecular Function; GO_CC: Gene Ontology Cellular Component; GO_BP: Gene Ontology Biological Process; KEGG: Kyoto Encyclopedia of Genes and Genomes.

**Figure 5 animals-10-01191-f005:**
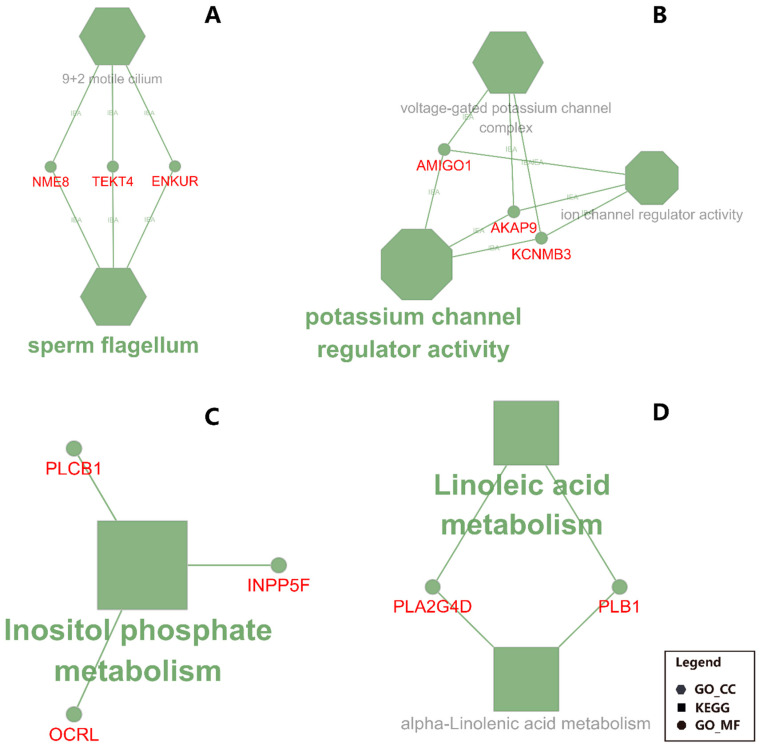
Functional annotation for liver and mammary in CS (corn stover) compared to MF (mixed forage) uncovered by ClueGO. Shown are the enriched terms in DEG between CS vs. MF. Particularly: (**A**) down-regulated DEG in liver; (**B**) up-regulated DEG in liver; **C**: down-regulated DEG in mammary tissue; (**D**) up-regulated DEG in mammary tissue. The size of the nodes reflects the statistical significance of each term. Larger the node size, smaller the P-value. The name of each group is given by the most significant term/pathway of the group. The nodes are grouped by similarity of their associated genes. The nodes in hexagon represents the terms of GO Cellular Component; the nodes in octagon represents the terms of GO Molecular Function; the nodes in rectangle represents the KEGG pathway. The small circles with symbols in red font represent the genes associated with the corresponding terms or pathways.

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
