# Peer review of "Diet Composition Affects Liver and Mammary Tissue Transcriptome in Primiparous Holstein Dairy Cows"

_animals, 2020, doi:10.3390/ani10071191_

Round 1

Reviewer 1 Report

This manuscript appears to describe a completely randomized study evaluating the effect of 2 completely different diets on the transcriptome of liver and mammary tissues of primiparous cows. The manuscript is publishable, however there are a number of aspects that require attention/clarification before the manuscript is acceptable for publication.

Here are my major concerns about this manuscript:

  1. In the title and throughout the manuscript the study is described as a comparison of diets containing different forage to concentrate ratios, which is true, but that effect is confounded by the forages used in the two different diets. In effect, the comparison is between different forages and different forage levels. I suggest the title be changed to something like “Diet composition affects liver and mammary tissue transcriptome in primiparous Holstein dairy cows”. Consequently, statements indicating F:C was the only difference among diets and the factor influencing results are incorrect and will need to restated.
  2. As stated above, the study described in this manuscript proposed to evaluate the effect of 2 diets on transcriptomes. I would expect to see the diets and composition presented in table(s) in this paper. The companion paper that is cited as containing that information was not published in this journal. I would like to see diet ingredients and diet composition, as well as the composition of any feeds that differed between the two treatment diets included in the manuscript.

Here are other minor suggestions:

  1. Page 1, line 16: change uppercase in Forage to lowercase
  2. Page 1, line 17: change “affect the milk” to “affect milk”
  3. Page 2, line 61: change “F:C=40:60 by mixed” to “F:C=40:60 with mixed”
  4. Page 2, line 68: add “the” between “in” and “liver”
  5. Page 2, lines 91 to 93: numbers are incorrect in the diet composition sentence. Please, check/adjust. Include table(s) with ingredients and composition
  6. Page 3, lines 96-98: please, consider adjusting the sentence beginning with “The total duration…”. The total duration of the study should be 11 weeks, with 2 weeks of adaptation period and 9 weeks of experimental period
  7. Page 3, Sections 2.4 and 2.5: both sections have the same title. Please, reconsider making it more specific
  8. Page 3, line 140: “Institute” should be capitalized
  9. Page 4, lines 152 and 156: is it “DAVID” or “David”?

Author Response

Response to Reviewer 1:

This manuscript appears to describe a completely randomized study evaluating the effect of 2 completely different diets on the transcriptome of liver and mammary tissues of primiparous cows. The manuscript is publishable, however there are a number of aspects that require attention/clarification before the manuscript is acceptable for publication.

Here are my major concerns about this manuscript:

  1. In the title and throughout the manuscript the study is described as a comparison of diets containing different forage to concentrate ratios, which is true, but that effect is confounded by the forages used in the two different diets. In effect, the comparison is between different forages and different forage levels. I suggest the title be changed to something like “Diet composition affects liver and mammary tissue transcriptome in primiparous Holstein dairy cows”. Consequently, statements indicating F:C was the only difference among diets and the factor influencing results are incorrect and will need to restated.

AU: Thanks for your suggestion. We changed the title as suggested. The F:C factor did combine with the factor forage source and cannot be separated. Thus, we accept your suggestion of emphasizing the comparison between two different diets (CS and MF). Accordingly, we rewrote the introduction and objective, and corrected relevant content in results, discussion and conclusion.

  1. As stated above, the study described in this manuscript proposed to evaluate the effect of 2 diets on transcriptomes. I would expect to see the diets and composition presented in table(s) in this paper. The companion paper that is cited as containing that information was not published in this journal. I would like to see diet ingredients and diet composition, as well as the composition of any feeds that differed between the two treatment diets included in the manuscript.

AU: Thanks for your suggestion. Following is the ingredients and chemical composition of experimental diets. Because this table has been published in our previous paper (Han et al. 2014), it will be redundant to add it in this paper. So, we would prefer to cite the original manuscript rather than to use it again. However, here is the table:

Table 1 Ingredients and chemical composition of experimental diets.

Experimental diets

CS (F:C = 40:60)*

MF (F:C = 60:40)

Ingredients [g/100 g of DM]

Corn stover

37.1

-

Alfalfa hay

-

28.4

Corn silage

-

26.5

Chinese wile rye

-

3.7

Corn

33.5

22.8

Wheat bran

3.0

-

Soybean meal

23.6

11.8

Cottonseed fuzzy

-

5.1

Calcium phosphate

0.4

0.6

Limestone

1.3

-

NaCl

0.5

0.5

Mineral-vitamin mix†

0.6

0.6

Chemical composition [g/100 g of DM]

Dry matter content

54.47

55.78

Crude protein

16.90

16.70

Neutral detergent fibre

41.01

44.18

Acid detergent fibre

21.15

26.06

Ether extract

1.58

2.24

Calcium

0.89

0.82

Phosphorus

0.21

0.31

Net energy lactation (NEL)# [MJ/kg of DM]

6.19

6.35

Notes: *F:C, forage-to-concentrate ratio; †Containing (per kilogram dry matter of premix): vitamin A 2,000,000 IU, vitamin D 600,000 IU, vitamin E 10,800 mg, ferrum 4080 mg, copper 4989 mg, zinc 180 mg, manganese 17,500 mg, iodine 180 mg; cobalt 8805 mg; #Estimated based on chemical compositions and 24 h gas production of diets (Hohenheim gas test).

Here are other minor suggestions:

  1. Page 1, line 16: change uppercase in Forage to lowercase

AU: We rewrote the first 2 sentences.

  1. Page 1, line 17: change “affect the milk” to “affect milk”

AU: Corrected in Page 1, line 16.

  1. Page 2, line 61: change “F:C=40:60 by mixed” to “F:C=40:60 with mixed”

AU: We rewrote the whole paragraph in page 2, line 57-68.

  1. Page 2, line 68: add “the” between “in” and “liver”

AU: We rewrote the whole paragraph in page 2, line 57-68.

  1. Page 2, lines 91 to 93: numbers are incorrect in the diet composition sentence. Please, check/adjust. Include table(s) with ingredients and composition

AU: Yes, the number of NEL was incorrect. It should be ‘NEL (±0.08 MCal/100 g of DM])’. We corrected it in page 2, line 82.

  1. Page 3, lines 96-98: please, consider adjusting the sentence beginning with “The total duration…”. The total duration of the study should be 11 weeks, with 2 weeks of adaptation period and 9 weeks of experimental period

AU: Corrected the sentence in page 2, line 86-88.

  1. Page 3, Sections 2.4 and 2.5: both sections have the same title. Please, reconsider making it more specific

AU: The titles of 2.4 and 2.5 were mistake and changed to ‘RNA Extraction and Microarray’ and ‘Statistical Analysis’ respectively in page 3, line 105 and page 3, line 127.

  1. Page 3, line 140: “Institute” should be capitalized

AU: Corrected in Page 3, line 130.

  1. Page 4, lines 152 and 156: is it “DAVID” or “David”?

AU: Should be DAVID. Corrected the ‘David’ in page 4, line 146 to ‘DAVID’. That sentence is needless. So we delete it.

Reviewer 2 Report

Theorethically, the objective of this manuscript was to evaluate the effect of the roughage to concentrate ratio on liver and mammary gland transcriptome in dairy cows. It is an interesting research work which is within the subject of the Animals journal. Nevertheless, unfortunately I cannot recommend its publication for the following reasons:
•        The main concern is related to the dissociation between the experimental design and the objetive.  To evaluate the effect of roughage to concentrate ratio it would have been convenient to minimize the effect of additional factors. In this study, at the same time as the roughage to concentrate ratio other factors change, such as the type of roughage and the composition of concentrate. Therefore, the effect of the diet on transcriptome cannot be associated only to roughage to concentrate ratio. Introduction and objetives should be profoundly modified accordingly.
•        On the other hand, in dairy cows diets, the proportion of concentrate is generally increased with the aim of increasing dietary nutrient density and intake. However, in the present study dry matter intake, nutrient intake and milk production were lower as the proportion of concentrate increased. It is an unusual response in dairy cows and it is not only consequence of a change in the roughage to concentrate ratio.
•        Interpretation of the results should probably be revised. For instance, authors stated that concentrate supplementation activates linoleate metabolims. Are the authors sure that linoleate was activated in CS cows and it was not downregulated in MF cows? It has been reported that C18 concentration in milk increases as dudoenal C18 supply increases, but to a certain level. From this level, the concentration in milk decreases as the duodenal supply increases. Please see Glaser et al. 2008. J. Dairy Sci.
•        Conclusion. Results cannot be interpreted in terms of the roughage to concentrate ratio and, therfore, conclusion neither. Likewise, autors must be sure wether a metabolic pathway was upregulated in one treatment or downregulated in the other treatment.

Author Response

Response to Reviewer 2:

Theorethically, the objective of this manuscript was to evaluate the effect of the roughage to concentrate ratio on liver and mammary gland transcriptome in dairy cows. It is an interesting research work which is within the subject of the Animals journal. Nevertheless, unfortunately I cannot recommend its publication for the following reasons:
•        The main concern is related to the dissociation between the experimental design and the objetive.  To evaluate the effect of roughage to concentrate ratio it would have been convenient to minimize the effect of additional factors. In this study, at the same time as the roughage to concentrate ratio other factors change, such as the type of roughage and the composition of concentrate. Therefore, the effect of the diet on transcriptome cannot be associated only to roughage to concentrate ratio. Introduction and objetives should be profoundly modified accordingly.

AU: Thanks for your suggestion. The F:C factor did combine with the factor forage source and cannot be separated. Thus, we accept your suggestion of emphasizing the comparison between two different diets (CS and MF). Accordingly, we rewrote the introduction and objective, and corrected relevant content in results, discussion and conclusion.

  •       On the other hand, in dairy cows diets, the proportion of concentrate is generally increased with the aim of increasing dietary nutrient density and intake. However, in the present study dry matter intake, nutrient intake and milk production were lower as the proportion of concentrate increased. It is an unusual response in dairy cows and it is not only consequence of a change in the roughage to concentrate ratio.

AU: Yes, the DMI indeed should be increased with the increase of concentrate. The decreased DMI in this study cannot be accounted only by the change of F:C. To an extent, it was likely induced by the lower palatability of corn stover. Therefore, we changed our focus to the CS diet per se and analyzed the effect of CS and MF diet on transcriptome of liver and mammary.

  •       Interpretation of the results should probably be revised. For instance, authors stated that concentrate supplementation activates linoleate metabolims. Are the authors sure that linoleate was activated in CS cows and it was not downregulated in MF cows? It has been reported that C18 concentration in milk increases as dudoenal C18 supply increases, but to a certain level. From this level, the concentration in milk decreases as the duodenal supply increases. Please see Glaser et al. 2008. J. Dairy Sci.

AU: Because this study involved two treatments and the differentially expressed genes in liver and mammary gland were analyzed between dairy cows fed CS diet and MF diet, the upregulated pathways in this study represent those pathways induced by CS diet compared with MF diet. Thus, data can only allow to see if pathways are induced or inhibited between CS and MF. In the present study we used MF as our control; thus, we assume this to be the steady-state and any change was related to MF. It is definitively possible that there was a decrease of linoleate metabolism in MF and not an increase in CS. Thanks for providing the reference above. We checked that study but we could not find the decrease the reviewer is pointing out above. According to the discussion section of the metanalysis study indicated (see “Percentage of 18:3” subsection)”: “The increase in milk 18:3 percentage was also very limited in absolute amount, except with protected supplements. Without protection, mean percentage in milk hardly reached 1.2% of total FA, compared with 0.67% for unsupplemented diets. For 18:2, only protection with formaldehyde seemed to protect dietary 18:3 somewhat from extensive rumen BH. Compared with basal levels, however, the increase in 18:3 percentage can be significant.”

  •       Conclusion. Results cannot be interpreted in terms of the roughage to concentrate ratio and, therfore, conclusion neither. Likewise, autors must be sure wether a metabolic pathway was upregulated in one treatment or downregulated in the other treatment.

AU: Thanks for your suggestion, we rewrote the introduction and objective, and corrected relevant content in results, discussion and conclusion accordingly. As mentioned above, we can only confirm that the induced or inhibited pathways were changed in CS compared with MF.

Round 2

Reviewer 1 Report

Critical concerns:

  1. Page 2, lines 81-83: present the average composition of the diets (DM, CP, NDF, and NEL). Current "deviation" (±) presented are meaningless for the reader, especially left unexplained.
  2. I understand that the inclusion of the table would be redundant, but the previous publication is even in a different journal. It would be a hardship on readers to search and subscribe to a separate journal to access basic, necessary information fundamental to understanding the results of your work. I strongly suggest Table 1 be included in the current manuscript or at least be included as supplementary information along with the paper.

Author Response

  1. Page 2, lines 81-83: present the average composition of the diets (DM, CP, NDF, and NEL). Current "deviation" (±) presented are meaningless for the reader, especially left unexplained.

AU: Thanks for your suggestion. We added the average composition of DM, CP, NDF, and NEL of the two diets and removed the deviation in Line 81-82.

  1. I understand that the inclusion of the table would be redundant, but the previous publication is even in a different journal. It would be a hardship on readers to search and subscribe to a separate journal to access basic, necessary information fundamental to understanding the results of your work. I strongly suggest Table 1 be included in the current manuscript or at least be included as supplementary information along with the paper.

AU: Thanks for your suggestion. We do need to consider the convenience of readers. We supplemented the table of ingredients and chemical composition of experimental diets in supplemental file 5.

Reviewer 2 Report

The authors have followed the recommendations and have answered point by point to all the questions asked. As a consequence, I consider the manuscript to be suitable for publication.

Author Response

Thanks a lot for your comments again!